# Hybrid Quantum Neural Network for Drug Response Prediction

**DOI:** 10.3390/cancers15102705

**Published:** 2023-05-10

**Authors:** Asel Sagingalieva, Mohammad Kordzanganeh, Nurbolat Kenbayev, Daria Kosichkina, Tatiana Tomashuk, Alexey Melnikov

**Affiliations:** Terra Quantum AG, Kornhausstrasse 25, 9000 St. Gallen, Switzerland

**Keywords:** precision oncology, drug response prediction, hybrid quantum machine learning, quantum computing in healthcare

## Abstract

**Simple Summary:**

This work successfully employs a novel approach in processing patient and drug data to predict the drug response for cancer patients. The approach uses a deep quantum computing circuit as part of a machine learning architecture to simultaneously consider the cell line and the chemical and predict its effect. The resultant hybrid quantum architecture predicted the drug response with 15% better effectiveness than its classical counterpart. This result presents a step towards designing personalized drugs using the abilities of quantum computers.

**Abstract:**

Cancer is one of the leading causes of death worldwide. It is caused by various genetic mutations, which makes every instance of the disease unique. Since chemotherapy can have extremely severe side effects, each patient requires a personalized treatment plan. Finding the dosages that maximize the beneficial effects of the drugs and minimize their adverse side effects is vital. Deep neural networks automate and improve drug selection. However, they require a lot of data to be trained on. Therefore, there is a need for machine-learning approaches that require less data. Hybrid quantum neural networks were shown to provide a potential advantage in problems where training data availability is limited. We propose a novel hybrid quantum neural network for drug response prediction based on a combination of convolutional, graph convolutional, and deep quantum neural layers of 8 qubits with 363 layers. We test our model on the reduced Genomics of Drug Sensitivity in Cancer dataset and show that the hybrid quantum model outperforms its classical analog by 15% in predicting ${\mathrm{IC}}_{50}$ drug effectiveness values. The proposed hybrid quantum machine learning model is a step towards deep quantum data-efficient algorithms with thousands of quantum gates for solving problems in personalized medicine, where data collection is a challenge.

## 1. Introduction

Cancer is a leading cause of death worldwide, accounting for nearly 10 million deaths in 2020, or nearly one in six deaths [1]. The main cause of death from cancer is widespread metastases formed due to the chaotic reproduction of corrupted cells. In any case of cancer, the normal regulation of cell division is disrupted due to a defect in one or more genes [2]. When the balance between old and new cells gets disrupted, a tumor, consisting of damaged cells with a propensity for unlimited cell division, develops. At some point, the presence of these cells begins to prevent the normal functioning of the cells and body tissues. The tumor becomes malignant or cancerous and spreads to nearby tissues. Usually, it takes many years for cells to build up enough damage to become cancerous. Sometimes, a defective gene is inherited from parents and in other cases, the mutation occurs due to the action of a toxic substance or penetrating radiation on the DNA of a single cell [3]. Each new case differs from the others due to the diversity of cell mutations, which gives us many treatment options. The selection of medicines is not easy for cancer, so the approach to treating each patient should be personalized [4]. The focus of personalized medicine in oncology is to focus on the most efficient and efficacious treatment.

Artificial intelligence can help develop personalized medicine by determining diagnoses and disease-appropriate therapy faster and less costly since machine learning algorithms have proven themselves capable of guaranteeing high accuracy of predictions, learning from patient data collected over years of clinical research [5]. The successes of artificial intelligence in the past decade have been numerous and fascinating. Machine learning models have been applied to problems varying in purpose and structure, from complicated object detection models [6] to impressive stable diffusion models [7]. A specific use case of machine learning with high potential yield is in the pharmaceutical industry, where innovations could dramatically affect people’s well-being. In this work, we specifically focus on the applications of machine learning algorithms to understanding and predicting the drug response of patients with tumors. It is worth noting that modern machine learning algorithms, namely deep learning models, solve problems better with lots of data since they can learn from a large dataset and extract structures and hidden dependencies from them. However, when it comes to personalized medicine, including predicting drug responses, collecting data is a challenge [8].

Quantum technologies can greatly help classical machine learning [9,10,11]. Since the existing classical models require significant computational resources, this limits their performance. Quantum and quantum-inspired computing models can potentially improve the training process of existing classical models [12,13,14,15,16,17,18], allowing for better target function prediction accuracy with fewer iterations. Some of these methods provide a polynomial speedup, critical for large and complex problems where small improvements can make a big difference. Among these advantages, quantum computing specifically demonstrates benefits over classical computing across various domains, including chemistry simulations [19,20] and within the medical and healthcare industry [21,22,23,24,25]. For example, in [26], it was shown that already existing machine learning algorithms, being quantum enhanced, significantly contribute to medical science and healthcare. The authors have also shown that the prediction of heart failure as the cause of death can be effectively and more precisely predicted with the help of quantum machine learning algorithms instead of classical ones. Among many existing quantum methods, quantum neural networks [27,28,29,30,31,32] are one of the most promising. For instance, in [33], the authors use a quantum generative adversarial neural network (GAN) with a hybrid generator to discover new drug molecules. The results of the article show that a classical GAN cannot properly learn molecule distribution. However, the proposed QGAN-HG (with only 15 extra quantum gate parameters) can learn molecular tasks much more effectively. In addition, hybrid quantum neural networks (HQNNs) can converge towards optimal solutions with fewer iterations and higher accuracy compared to classical neural networks, particularly when dealing with small datasets [34]. The previous examples show that HQNNs can have an advantage over purely classical approaches. They also indicate that quantum neural networks have the potential to solve pharmacological problems [35], such as the task of predicting the response of patients to different medications.

In this article, we propose an HQNN to solve the problem of predicting ${\mathrm{IC}}_{50}$ values. The dataset we use is called GDSC and contains information on mutations in genomes that can be matched with different types of cancer, as well as the chemical formula of drug molecules. We aim to predict how the mutation will respond to the drug, how effective the drug will be and the value of ${\mathrm{IC}}_{50}$. Section 3.1 details the dataset. Section 3.2 describes in detail what preprocessing we perform and the architecture of the network used to solve the ${\mathrm{IC}}_{50}$ prediction task, which is a combination of classical and quantum layers. The classical part of the HQNN consists of several subnets: graph convolutional networks and convolutional networks for extracting information about drugs and cancer cell lines, respectively. The quantum layer of 8 qubits that we first introduce in this work is called the Quantum Depth-Infused Neural Network Layer. This layer consists of an encoding layer, a Variational Quantum Circuit of 363 layers, and a measurement layer. The result of measuring one qubit is a classical bit, the drug response prediction value for a drug/cell line pair. Section 3.3 describes the training process and compares the HQNN and a classical one where the quantum layer is replaced by classical fully connected (FC) layers. The hybrid quantum approach showed a 15% better result in predicting the ${\mathrm{IC}}_{50}$ values compared to its classical counterpart.

## 2. Machine Learning for Drug Response Prediction

In this section, we showcase existing Machine Learning approaches that led up to this paper. All the models mentioned below had experiments set using the Genomics of Drug Sensitivity in Cancer (GDSC) [36] dataset, including the drug-specific description, cancer cells, and calculated ${\mathrm{IC}}_{50}$ value.

The ${\mathrm{IC}}_{50}$ metric is an important measure of drug response used to evaluate the effectiveness of therapy against different types of cancer. It represents the half-maximal inhibitory concentration. The ${\mathrm{IC}}_{50}$ of a drug can be determined by constructing a dose-response curve while examining the effect of different drug concentrations on the disease response. The larger the ${\mathrm{IC}}_{50}$ value is, the higher the drug concentration needed to destroy cells by 50%. In other words, the drug is less effective, and the response of the disease to this medicine is low. Similarly, the smaller the ${\mathrm{IC}}_{50}$ value is, the lower the concentration of the drug required to suppress cells by 50% and the more effective the drug is.

Among the first approaches to solving the problem of predicting drugs using sensitivity to cancer cells was the one implemented in 2012 [37]. Using the elastic net model, it was possible to identify complex biomarkers and match them with sensitive cancer cells. The elastic net model uses coordinate descent to predict with a linear model and uses the determination coefficient to set the parameters for the estimator. Elastic Net adopted genomic features to reproduce the tumor environment and thus considered only part of the features, which was rather small to reach high accuracy and universality concerning the larger set of drugs and diseases.

Random Forest is a classification model that applies a sequence of queries to a set of data samples to linearly restrict and split it into classes [38]. It was shown that the model considered cancer cells and drug features performed well as multi-drugs yet left space for increasing the model’s robustness by considering more input features of the cell lines.

Ridge regression, in a similar way to linear regression, follows a path of coordinate descent. It then changes the cost function in a way so that it could regain the regression model’s stability and avoid overfitting. A model with an additional quadratic term in its loss function [39] and that consisted of baseline gene expression only managed to perform better than several existing biomarkers, though the model is believed to be improved by being more specific in expressing other levels of gene structure.

When a Convolutional Neural Network gets a picture as an input, it assigns weights to a filter and does convolution to an existing layer of pixels. A filter or kernel extracts the dominating features and the pooling layer is implemented to reduce the object’s volume. After going through the series of convolutional and pooling layers, the resulting features are processed by the FC layers and then distributed into classes. The proposed approach [40] employed an ensemble of five models that gave an average predicted ${\mathrm{IC}}_{50}$ value for cancer cell lines (instead of cancer types). Along with genomic fingerprints, the models used selected mutation positions.

Since the first deep learning models emerged, their application in personalized medicine increased with their development. In 2019, a deep learning multi-layered model was proposed, starting with an input layer, then nonlinearity layers and a classification layer at the end [41]. The pathways to each drug were analyzed to increase the efficiency of the neural net performance. The accuracy of the predictions increases as the training datasets become larger, so it is believed that it is possible to improve the model with larger datasets.

Another deep learning-based model applies the method of late integration of mutation, aberration number and omics-data, DL MOLI [42]. The model preprocesses the data to send the result through three encoder neural sub-networks to extract features. Then, the model connects the features into one representation of the given tumor sample and follows the classifier structure with the cost function considering triplet loss to separate responder samples from non-responders.

Visual neural networks applied to the given cluster of problems represent human cancer cell models. In this model, considering both drug and genotype features, their embedding was performed in parallel [43]. The drug branch was built as an artificial network of three hidden FC layers, where each drug was described as a fingerprint at the input. Genotypes were binary encoded and passed through the network, which was constructed as cell subsystems, and each was assigned its number of neurons.

In the presented variety of approaches to solving the problem of personalized medicine, each classical neural network tried to integrate data into the model and, in its way, tried to reproduce the conditions of real life as best as possible. However, until now, previous approaches not only did not take into account the possible factors influencing the study to the maximum, but also an architecture capable of taking into account the influence of all these factors could not be built, leaving alone the problem of small datasets and overfitting. Consequently, the result was given for some special cases.

In this article, we present a method for solving the problem of predicting the ${\mathrm{IC}}_{50}$ values using an HQNN, as seen in Figure 1. To date, we have not found studies that would solve the problem of predicting the ${\mathrm{IC}}_{50}$ values using quantum or HQNNs. Our approach solves the drug response prediction problem more accurately than its classical counterpart on a small dataset. In the next section, the hybrid quantum method and results will be described in detail.

## 3. Results

This paper explores the use of HQNNs in drug response prediction. The proposed model is a combination of graph convolutional, convolutional, and quantum layers. We tested our model on a small part of the dataset using the GDSC database (Section 3.1). HQNNs can provide higher learning efficiency, requiring fewer iterations, and can show lower prediction error in the ${\mathrm{IC}}_{50}$ values by 15% compared to the classical analog, as will be shown below.

### 3.1. Description of Dataset

The dataset known as the GDSC [36] is considered to be the largest database keeping information about cancer cell line sensitivity on prescribed anti-cancer drugs and genetic correlations.

The GDSC database consists of cell line drug sensitivity data generated from ongoing high-throughput screening, genomic datasets for cell lines and the analysis of genomic features, or systematic integration of large-scale genomic and drug sensitivity datasets. For example, see Figure 1. The compounds selected for the screening of cancer are anticancer therapeutics. They comprise approved drugs used in the clinic, drugs undergoing clinical development, drugs in clinical trials, and tool compounds in early-phase development. They cover a wide range of targets and processes implicated in cancer biology. Cell line drug sensitivity is measured using fluorescence-based cell viability assays following 72 h of drug treatment.

Values from the dataset include the ${\mathrm{IC}}_{50}$ value, the slope of the dose-response curve, and the area under the curve for each experiment. We used ${\mathrm{IC}}_{50}$ as the primary because it is a widely accepted measure of drug response prediction tasks. However, we believe that our HQNN model can be adapted to predict AUC values as well. While we have not specifically tested our model using the AUC metric, the general architecture of our HQNN can be applied to any output, making it a versatile tool for drug response prediction, although some modifications to the loss function may be necessary. It is worth noticing that mRNA expression profiles are also important characteristics of cell lines and can be used to represent them. The purpose of this particular study, however, was to find out how well the model can predict IC50 values for cell lines. In this study, we used version 6.0 of the dataset.

### 3.2. Hybrid Quantum Neural Network Architecture

The principle of operation of the neural network, as shown in Figure 1, is the distribution of drugs, represented in the form of a chemical formula in the dataset, and cancer cells encoded in a binary chain over two parallel working neural networks and all the results placed in a quantum neural network layer for analysis and prediction of the ${\mathrm{IC}}_{50}$ value.

In this section, we present a solution to the problem of ${\mathrm{IC}}_{50}$ prediction and describe in detail the network architecture used. The architecture of an HQNN is illustrated in Figure 2 and consists of three sub-networks: a neural network for cell line representations (Section 3.2.1), a neural network for drug representations (Section 3.2.2), and a quantum neural network (Section 3.2.3). It is worth noting that we were inspired to use such a classical part of our HQNN [44].

#### 3.2.1. Cell Line Representation

In biology, a cell line is a population of human cells that normally would not multiply indefinitely; however, because of the gene mutation, these cells avoided cellular ageing and continue to divide infinitely. Thus, in our studies, the cell line represents the tumor itself as a binary vector, where 1 indicates the presence of genomic mutation and 0 is its absence, as shown in Figure 2. A one-dimensional convolutional layer was used to encode the cell line’s genomic features, represented as one-hot encoding, namely a 735-dimensional binary vector. Then, the feature map was passed through the Max Pooling layer to reduce the spatial volume of the map. After three repetitions of the convolutional layers, the final FC layer flattens the output into a vector of 128 dimensions.

#### 3.2.2. Drug Representation

The drug is represented as a graph in the format of SMILES, using the PubChem library [36]. In this format, the drug is written in a line. Each node of the graph contains information describing its graphical and chemical representation, including the atom type, the chemical bonds, the branching of the molecule, its configuration, and its isotopes. The atom type is encoded with the standard abbreviation of the chemical elements, the bonds are represented with special symbols, branches are described with parentheses, and isotopes are specified with a number equal to the integer isotopic mass preceding the atomic symbol. Then, the drug data is transformed into a molecular graph using RDKit software [45]. Each of the 223 drugs has its unique chemical structure, which can be naturally represented as a graph where the vertices and edges denote chemical atoms and bonds, respectively. Thus, each graph obtained from one line of SMILES contains information about one specific drug and has a number of vertices equal to the number of atoms in the molecule of this drug, and the edges in the graph are responsible for the bonds between atoms in the molecule. Representing drugs in graphs is more suitable than in strings since it conserves the nature of the chemical structures of drugs. Since drugs were represented in graphs, a Graph Convolutional Network [46] was used to learn its features.

A Graph Convolutional Layer was taken with 2 matrices as an input: a feature matrix $X\in R^{N\times F}$ and an adjacency matrix $A\in R^{N\times N}$, which displays the connections between atoms, where *N* is the number of atoms in a molecule, *F* is the atomic features of a 78-dimensional binary feature vector, and then a node-level output with *C* features for each node is produced.

In the used sub-network there were three graph convolutional layers. After each, an activation function ReLU [47] was used. After the last graph convolutional layer, a global max pooling was applied. The obtained output was turned into a 128-dimensional vector by the sequence of two FC layers with sizes 312, 1024, and 128, respectively.

#### 3.2.3. Quantum Neural Network

The resulting combination of cell and drug data, constituting a vector of 256 nodes, was transformed into a quantum layer of the HQNN, composed of three parts: embedding, variational layers, and measurement. In this work, we employ a large and deep quantum neural network. This size arises from a need to propagate uncompressed data (256 neurons) through the classical preparation networks to the quantum layer. Commonly used quantum neural network architectures normally encode each feature on a separate qubit [34,48,49], but for 256 neurons, this is not accessible to today’s quantum engineers for two main reasons: (1) this qubit count is far beyond the fault-tolerant readily accessible quantum simulator or hardware, and (2) the noise-free barren plateau problem [29] affects this region with high potency, rendering the model impossible to train. Instead, we take inspiration from the data re-uploading method introduced in [50] and developed further in [51] to create a lattice of features: we use 8 qubits, and create a lattice of length 32 (blue square in Figure 1).

Each of the 256 features leading up to the quantum layer is encoded on this lattice in such a way that the first 8 features ${\left\{\phi_{i}\right\}}_{i=1}^{8}$ are encoded on the first lattice length, features 9–16 ${\left\{\phi_{i}\right\}}_{i=9}^{16}$ on the second length, and so on. To encode these classical features into the quantum Hilbert space, we use the “angle embedding” method, which rotates each qubit in the ground state around the *Z*-axis on the Bloch sphere [52] by an angle proportional to the corresponding value in the input vector. This operation encodes the input vector into the quantum space, and the resulting quantum state represents the input data from the previous classical layer. The names, descriptions, matrices, and representations of all used quantum gates are presented in Table 1.

To ensure the highest Fourier accessibility [51] to the feature encoding, entangling variational layers (purple square) are placed between each layer of feature encoding. Each variational layer consists of two parts: rotations with trainable parameters, and control gates, which are typically subsequent CNOT operations [53]. The rotations serve as quantum gates that transform the encoded input data according to the variational parameters, while the CNOT operations are used to entangle the qubits in the quantum layer, allowing for the creation of quantum superposition. The number of variational layers in each lattice length, in each blue square, equals five, but it is important to note that the variational parameters for each variational layer are different in each of the five repetitions. Moreover, we apply five variational layers (green square) for greater representativeness of the model before all encoding layers. Thus, the total number of weights in the quantum part of the HQNN is calculated as 8×5+8×5×32 and equals 1320.

In the measurement part, all qubits but the first make a CNOT operation on the first qubit so that the Z-measurement is propagated to all qubits. The *Z*-measurement is then augmented using a multiplicative and an additive trainable parameter to fit the requirements of the ${\mathrm{IC}}_{50}$ value. Thus, the output of the HQNN is the prediction of the ${\mathrm{IC}}_{50}$ value of the particular chemical/cell line pair.

It is noteworthy that a quantum circuit of this depth would be far too difficult a challenge for today’s noisy quantum devices and it is likely that this architecture in its current form will stay on a quantum simulator for the foreseeable future. However, it is possible to create mathematically identical circuits with much shallower depth but higher qubit counts. This means that by increasing the number of qubits, we get the same model but much shallower and thus with a higher potential for being run on a noisy quantum device.

The HQNN was compared with its classical counterpart, the architecture of which differs only in the last part: instead of a quantum layer, the classical network has two FC layers, the numbers of neurons which are eight and one. In terms of the number of parameters, these two models also differ only in the last part: the HQNN has 1320 variational parameters, while the classical model has 2056.

### 3.3. Training and Results

In our experiment, the HQNN was evaluated on drug/cell line pairs provided by the GDSC database with known ${\mathrm{IC}}_{50}$ values. The used data consisted of 223 × 948 = 172,114 pairs of drugs and cell lines. The corresponding received response values were normalized in the range of (0, 1). We used a logistic-like function: for each ${\mathrm{IC}}_{50}$ value *y*, we applied the following formula: $\mathrm{norm}\left(y\right)=1/\left(1+y^{0.1}\right)$, where y>0. The observed/expected ${\mathrm{IC}}_{50}$ value, which has to be a positive number greater than zero [38]. The preprocessed data (172,114 pairs) were shuffled and distributed once into training (80%) and testing (20%) sets. Since it is difficult to collect data in tasks related to the pharmaceutical industry and personal medicine, and HQNNs performed well in solving problems on a small dataset, the dataset was reduced to 5000 samples for training and 1000 for testing procedure. We chose this split ratio to ensure enough data to train our model while still reserving a reasonable amount for testing. The split was performed randomly. To increase the representativeness of the dataset and prevent accidental skewing, the order of samples in the training set is shuffled at each epoch, while in the testing set it remains the same. As an optimizer, Adam [54] was used with a learning rate equal to $1.8\times 10^{-3}$.

In our study, the hyperparameters were determined through a combination of trial and error and a grid search approach. For the number of qubits, we experimented with different numbers of qubits for each encoding length until we found a balance between model accuracy and computational resources. For the learning rate, we tried different values within a reasonable range to find the optimal value that minimized the loss function during training. Other hyperparameters, such as the number of variational layers and the type of optimizer, were also optimized using a grid search approach. We did not use a separate dataset in our work for the hyperparameter optimization method, we trained our models on the training set and then evaluated them on the testing set. Finally, we selected the hyperparameters that gave the best performance on the testing set.

To evaluate the effectiveness of the model, we used the mean square error (MSE) metric, which measures the discrepancy between the predicted ${\mathrm{IC}}_{50}$ value ($Y_{i}$) and the corresponding ground-truth value ($O_{i}$) for each drug/cell line pair (*i*) in our dataset. The MSE is defined as follows:$$\mathrm{MSE}=\frac{1}{N}\sum_{i=1}^{N}{\left(O_{i}-Y_{i}\right)}^{2}$$
where *N* denotes the number of drug/cell line pairs in the dataset. The successful performance of the model is determined by small values of the MSE—the lower, the better.

All machine learning simulations were carried out in the QMware cloud [34,55], on which the classical part was implemented with the PyTorch library [56] and the quantum part was implemented with the PennyLane framework. PennyLane offers a variety of qubit devices. We used the lightning.qubit device, which implements a high-performance C++ backend. To obtain the gradients of the loss function with respect to each of the parameters, for the classical part of our HQNN, the standard back propagation algorithm [57] was used and for the quantum part, the adjoint method [58,59] was used. The results of the simulations are shown in Figure 3. We observe that the MSE loss function of the HQNN decreases monotonically during training, while the classical model exhibits a volatile behavior. This indicates that the HQNN is more stable and efficient in minimizing the loss function.

This is due to the inherent properties of quantum computing, such as superposition and entanglement, which can enhance the ability of the model to learn complex relationships and patterns in the data. These properties can also help avoid overfitting and improve generalization, leading to better performance on unseen data. Therefore, by utilizing quantum computing in machine learning, we can potentially achieve more robust and reliable models that can better understand complex systems, such as drug response prediction.

Thus, the network with the deep quantum layer demonstrates a better result than the classical one, and the improvement in the loss of the HQNN is more than 15% compared to the classical model with the same architecture.

We also conducted an experiment by training the hybrid and classical models with the same hyperparameters on a different amount of training data: on 50, 200, and 5000 numbers of samples. The results are presented in Figure 4. The *Y*-axis shows the difference in the MSE loss on the testing set, which consisted of 10, 50, and 1000 data points, respectively, to maintain the training/testing ratio, between the hybrid and classical models. As one can see from the graph, the fewer training data, the greater the difference in loss, and the better the quantum model is compared to the classical one. This suggests that HQNNs are particularly useful for tasks such as personalized medicine, where obtaining large amounts of data may be challenging, and where classical neural networks may encounter issues such as overfitting or underfitting. Our findings provide evidence that our method can be applied to datasets with small sample sizes and still achieve good performance.

## 4. Conclusions

This study presents a novel HQNN consisting of a deep quantum circuit with 1320 trainable gates for anticancer drug response prediction. In our model, drugs were represented as molecular graphs, while cell lines were represented as binary vectors. To learn the features of drugs and cell lines, graph convolutional and convolutional layers with FC layers were used. Then, a deep quantum neural network predicted the response values for the drug/cell line pairs. The deep quantum neural network is a newly designed layer called the Quantum Depth-Infused Neural Network Layer. We were able to encode information from 256 neurons into just 8 qubits. As far as we know, this is the first work in which HQNNs are used to predict the value of ${\mathrm{IC}}_{50}$.

In addition, we have demonstrated that our HQNN is 15% better than its classical counterpart, in which the quantum layer is replaced by an FC classical layer with eight neurons. We tested our models on a reduced dataset presented by GDSC consisting of 5000 and 1000 training and test drugs/cell line pairs, respectively. Moreover, we have shown that the quantum model outperforms the classical model more, especially with little training data.

It is important to note that the small sample problem in drug response prediction is not only about the number of samples but also the complexity and heterogeneity of the dataset. The GDSC dataset contains more than 1000 different cancer cell lines, each with its own unique genetic mutations and characteristics, and hundreds of different drugs. The number of possible drug-cell line combinations is therefore much larger than the number of samples we used in this study. In this context, our dataset can be considered relatively small. Furthermore, it is worth noting that the number of available samples for some drug-cell line combinations in the GDSC dataset is still limited. For example, some combinations may only have a few data points available, making it challenging to accurately predict drug response. Therefore, our method is still relevant in addressing the small sample problem in drug response prediction, even though we used a dataset that is relatively large compared to some other studies. Additionally, our method has the potential to uncover important molecular mechanisms underlying drug response, which is crucial for the development of personalized medicine where data collection is a difficult task. In addition, in anticancer drug design and understanding cancer biology, identifying cell lines that are important for drug sensitivity or resistance.

With the ability to predict the ${\mathrm{IC}}_{50}$ values of a chemical/cell line pair, researchers can more easily identify potential candidates for drug development, prioritize experiments, and reduce the need for costly and time-consuming laboratory experiments. Moreover, the use of a HQNN allows for the exploration of complex relationships between features that may not be easily identifiable using classical methods, potentially leading to new insights and understandings of cancer biology, and also to the repurposing of existing drugs for cancer treatment.

In future work, we are planning to calculate a list of the ${\mathrm{IC}}_{50}$ values for certain drug/cell line pairs using the models produced in this work, as well as increase the complexity and performance of the HQNN. It would also be valuable to test this architecture on other cancer-related datasets as we would like to assess the applicability of the HQNN to the wider healthcare context.

## Figures and Tables

**Figure 1 cancers-15-02705-f001:**
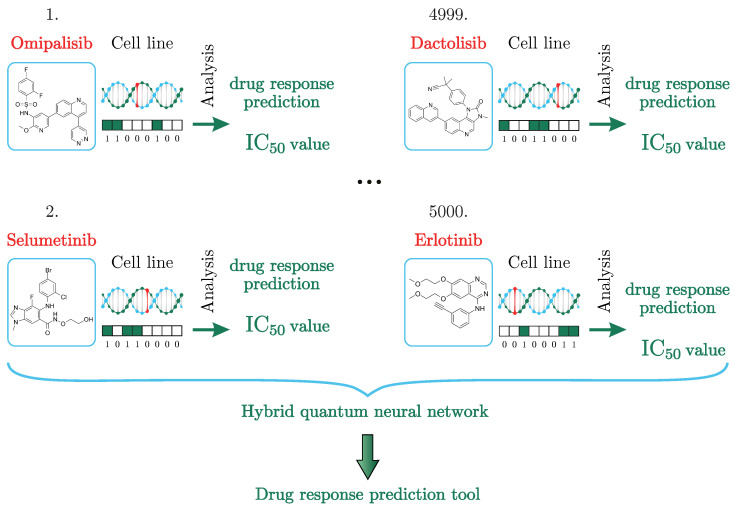
In each case, the drug response is calculated for a pair of chemical formulas of the medication/cell line. The drug response is evaluated using the ${\mathrm{IC}}_{50}$ value. The machine learning algorithm is a powerful drug predicting tool as it saves effort and time that could have been spent on tests in the laboratory and waiting for the results of experimental therapy courses to check whether there are improvements in the patient’s condition.

**Figure 2 cancers-15-02705-f002:**
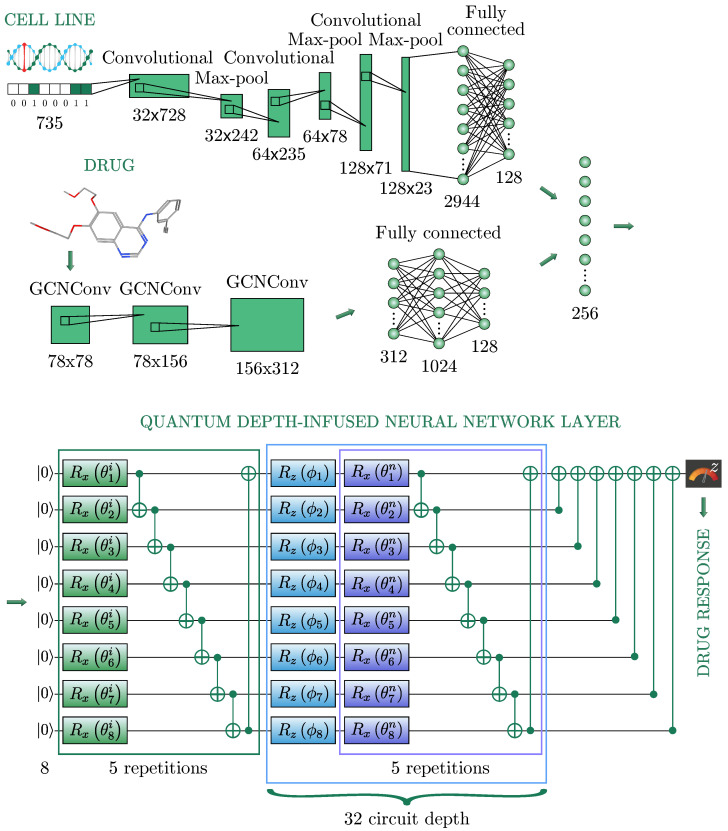
HQNN architecture. The cell line is fed to the network as a vector with 735 elements to a one-dimensional convolutional network. In parallel, the chemical composition of the drug is passed to a graph convolutional neural network. Each of these two is then processed in parallel according to the graphic above, to reach an FC layer of 128 neurons. Then, they are combined into a single layer of 256 (2 × 128) neurons. Classical information from every 8 classical neurons of 256 neurons is passed to the Quantum Depth-Infused Neural Network Layer. The information is encoded into rotation angles ${\left\{\phi_{j}\right\}}_{j=1}^{256}$ around the *Z*-axis into each of the 8 qubits of the 32 quantum circuits forming a quantum layer. The total number of variational parameters ${\left\{\theta_{k}^{h}\right\}}_{k=1}^{8}$ is 1320. The measurement result of the first qubit predicts the ${\mathrm{IC}}_{50}$ value.

**Figure 3 cancers-15-02705-f003:**
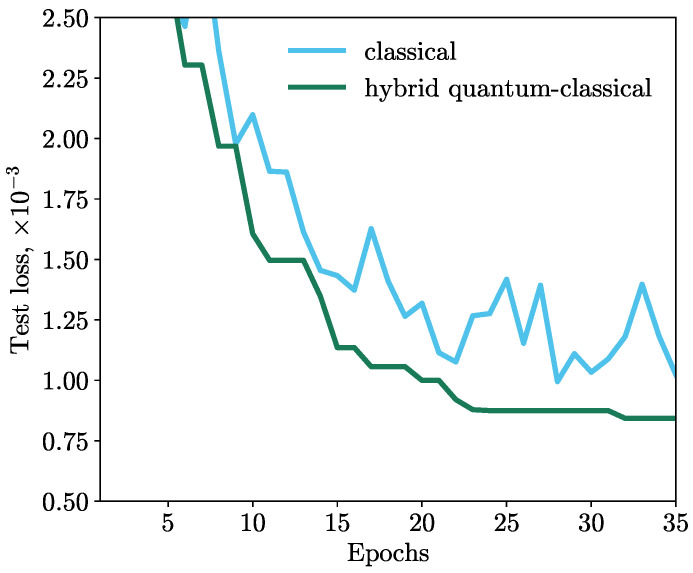
Dependence of the MSE loss on the number of epochs. The HQNN outperforms the classical counterpart by 15%.

**Figure 4 cancers-15-02705-f004:**
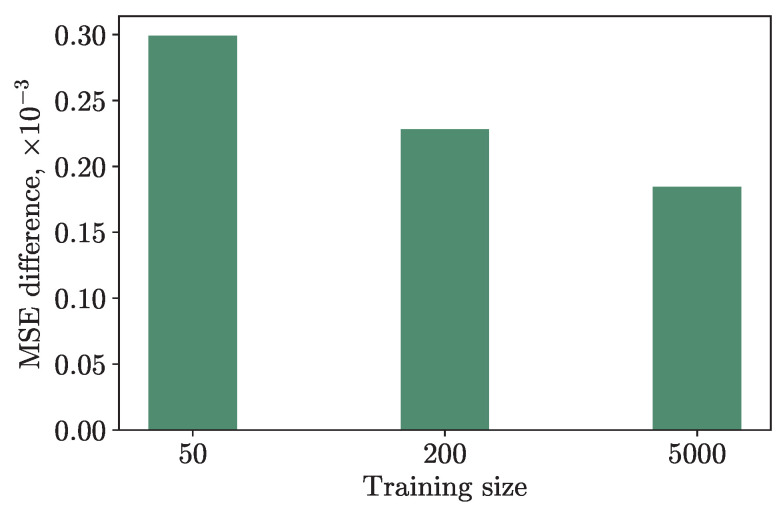
Dependence of the difference between MSE losses of the classical and hybrid model on training sizes. The fewer training data, the greater the difference in loss, and the better the quantum model is compared to the classical one.

**Table 1 cancers-15-02705-t001:** Quantum logic gates.

Action	Elements Name	Elements Description	Matrix Representation	Elements Notation
1	Rotation operator *X*	X(θ)=exp−iσxθ/2	cos(θ/2)−isin(θ/2)−isin(θ/2)cos(θ/2)	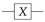
2	Rotation operator *Y*	Y(θ)=exp−iσyθ/2	cos(θ/2)−sin(θ/2)sin(θ/2)cos(θ/2)	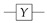
3	Rotation operator *Z*	Z(θ)=exp−iσzθ/2	exp(−iθ/2)00exp(iθ/2)	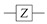
4	Controlled NOT gate	CNOT =expiπ4I−σz1I−σx2	1000010000010010	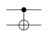

## Data Availability

This work uses the Genomics of Drug Sensitivity in Cancer dataset [36] for the studies.

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
