# Peer review of "Hybrid Quantum Neural Network for Drug Response Prediction"

_cancers, 2023, doi:10.3390/cancers15102705_

Round 1
Reviewer 1 Report
In this work, the authors proposed a hybrid quantum neural network for drug response prediction, which was tested on GDSC dataset to predict IC50 values. The method was shown to improve the prediction (measured by MSE) by 15% than the classical counterpart. The article is well written!
11. The authors stress that QNN is especially suited to solve the small sample problem. But the experiment in this study used 5000 samples for training and 1000 for testing procedure. This is not the typical “small sample problem” that we have in reality.
22. Besides the “definition” of small sample dataset, I also don’t see the advantage of this method in solving the small sample problem. Maybe the authors can do an experiment to show in order to achieve the same MSE, HQNN needs few training samples than the classical counterpart?
33. The training and testing procedure is still unclear to me: whether the split is done once or multiple time, or cross-validation, and whether the samples were randomly split into the training and testing.
44. Can the authors comment on the choice of IC50 instead of AUC? Will this model work on AUC as well?
55. This model uses cell line mutation profile as the representation of the cell lines. What about mRNA expression profile which is also an important characteristic?
66. In Fig.3, are the loss calculated from the same training data (after the shuffling) in each epoch for the classical and HQNN models? My question is about whether the loss is compared in a fair way.
77. In Fig.3, can the authors give explanations why the MSE decreases monotonically using the HQNN model while the classical one gives “volatile” MSE?
88. How are the hyperparameters determined, including the number of qubits, learning rate etc.?
99. Can the author shed light on how their method can help in understanding cancer biology?
110. Typo: “The HQNN was compared with its classical counterpart, the architecture of differs only in the last part”
111. Typo: “the dataset was redused to 5000 samples for training”
Reviewer 2 Report
The authors present a new hybrid quantum neural network (HQNN) to predict IC50 values of chemotherapy drugs from the Genomic of Drug Sensitivity in Cancer (GDSC) dataset. The main innovation in this manuscript is the incorporation of a quantum neural network layer into a classical machine learning (ML) neural network. The manuscript mainly concentrates on describing the new HQNN. The only net result presented from the HQNN’s utilization is that this network outperforms its classical analog by 15% in the prediction of IC50 values of drugs after ~30 epochs; this is a significant result. The representation of the chemotherapy drugs in terms of graph theory and other mathematical devices is interesting, but this and other classical ML features have precedents in the cited literature. The authors also present an appropriate review of previous classical ML approaches for drug response prediction. With one exception noted in number 2 below, the authors describe their HQNN well. Tables and pictures are properly presented. However, please note that the submitted manuscript is not in the standard Cancers template.
The new HQNN can make a positive impact in chemotherapy, and from that point of view, its publication will be relevant for the Cancers’ readership. My only concern is the sparsity of results demonstrating the accuracy and efficiency of the HQNN; however, the introduction of this neural network with only one key result may be acceptable due to its novelty. This manuscript can be published in Cancers if the authors properly address the following concerns:
1. The grammar and style of the written English in the manuscript is good, with only a few idiosyncratic expressions not impeding its understanding. There are a few typos in the manuscript (e.g., on page 2, “optima” could be replaced with a more precise word choice; in the caption of Fig. 2, “according the schematic above” should be “according to the graphic above”, etc.): the authors should correct them. In addition, some acronyms (e.g., GDSC) are not defined at their first appearance but later, while others are defined more than once.
2. Can the authors explain in the text in some detail the working of the quantum computing circuit of the quantum neural network layer? In other words, can they explain how the x and z rotational gates and the control gates process the data from the beginning to the end in more detail? Some readers will not be able to understand this key part of the manuscript without further elucidation.
Round 2
Reviewer 1 Report
Thank the authors for their extensive reply. The quality of the paper is improved by addressing the issues. A few remaining points to be addressed to convince the readers and to increase the significance and impact of their work:
1. Maybe the authors can give an overview graph (e.g bar graph) to show the sample sizes for all combinations in the training set, together with the prediction performance for that combination (if it’s in the test set as well). In this way, we can see whether a small training set for a particular combination still gives a good prediction in the test set. Furthermore, a comparison needs to be done against classical counterpart, and these prediction results can be included in the same plot.
2. Can the authors specify how they normalized the IC50 values into the range of (0,1)? Linear scaling using MinMax or so?
3. About whether this model also works on AUC, it would be better if the authors’ response to the comment is added into the manuscript to increase the significance and impact of their work.
4. About the choice that mRNA is not used to represent the cell lines, it would be better if the authors’ response is added into the manuscript to specify the scope and aim of their work, since many researchers would expect the mRNA profiles to be used as the input.
5. About the parameter optimization, the authors need to specify on which dataset the optimization was performed. What is the “validation set”? Is it within the training set? This needs to be specified.
Reviewer 2 Report
Please, read attached file.
